# The Scent Enriched Primate

**DOI:** 10.3390/ani13101617

**Published:** 2023-05-12

**Authors:** Emily J. Elwell, Stefano Vaglio

**Affiliations:** 1Animal Behaviour & Wildlife Conservation Group, School of Life Sciences, University of Wolverhampton, Wolverhampton WV1 1LY, UK; E.J.Elwell@wlv.ac.uk; 2University College—The Castle, Durham University, Durham DH1 3RW, UK

**Keywords:** behavioural indicators of stress, physiological indicators of stress, captive breeding, captive welfare, scent enrichment, zoo animals

## Abstract

**Simple Summary:**

In the face of a global biodiversity decline, zoos worldwide play important roles in conservation via efforts such as providing breeding programmes and reintroductions into the wild. Zoo populations are important as a buffer against extinction, but substantial differences between the wild and zoo environments can lead to health issues. These problems, in turn, can impact the reproductive success of individuals. Consequently, some primate species have reduced breeding success when housed in zoos compared to their wild counterparts. To prevent the onset of behavioural, physiological, and cognitive negative effects and to continually improve the welfare of their animals, zoos widely implement different types of environmental enrichment. Enrichment can take many forms, but sensory enrichments are less studied. Scent enrichments are less utilized despite multiple research studies showing that they may affect positively the well-being of zoo-housed animals, including non-human primates. Despite being traditionally considered to have a poor sense of smell, more evidence is emerging to suggest that olfaction is important in primates. This review article therefore focuses on scent-based enrichment and the specifics of primate olfaction.

**Abstract:**

Zoos worldwide play an important role in both in situ and ex situ conservation via efforts such as providing breeding programmes and reintroductions into the wild. Zoo populations are crucial as a buffer against extinction. However, a mismatch between the wild and zoo environments can lead to psychological as well as physiological health issues, such as stress, boredom, diabetes, and obesity. These problems, in turn, can impact the reproductive success of individuals. Consequently, some primate species have reduced breeding success when housed in zoos compared to their wild counterparts. To prevent the onset of behavioural, physiological, and cognitive negative effects and to continually improve the welfare of their animals, zoos widely implement different types of environmental enrichment. There are many forms enrichment can take, such as feeding, puzzles and training, but sensory enrichments, including implementing the use of scents, are currently understudied. Scent enrichments are less utilized despite multiple research studies showing that they may have positive effects on welfare for zoo-housed animal species, including non-human primates. Despite being traditionally considered to be microsmatic, various lines of evidence suggest that olfaction plays a larger role in primates than previously thought. This review therefore focuses on scent-based enrichment and the specifics of captive primates.

## 1. Introduction

Biodiversity has declined significantly over the last few decades. Captive conservation breeding is, therefore, becoming a very important tool for conservationists [1,2]. In this regard, several factors have been identified that are likely to contribute to the breeding success of captive animals, including husbandry practices such as diet and the use of environmental enrichment [2]. However, the relationship between environmental enrichment and reproductive success is still poorly understood and has received little focus thus far. Nevertheless, as enrichment aims to improve both the physiological and psychological status of individuals, it should have some impact upon reproduction.

In this paper, we review the link between captive conservation breeding and environmental enrichment programmes in zoos, the background information about environmental enrichment as a whole and then the knowledge specifically regarding olfactory enrichment. Particularly, we explore the case of non-human primates and the potential of scent enrichment to increase the reproductive success of zoo-housed primates.

Specifically, this review article examines the following:Environmental enrichment and conservation breeding in zoos.Olfactory communication and reproduction in captive primates.Scent-based enrichment and zoo-housed primates.

## 2. Environmental Enrichment and Conservation Breeding in Zoos

### 2.1. Environmental Enrichment

Environmental enrichment is widely implemented across zoos to improve welfare and prevent the onset of, or mitigate, the effects of stereotypic or negative behaviours [3]. “Welfare” is a very broad concept that incorporates the physical and psychological health of animals and can be measured [4], but sometimes it includes immeasurable aspects (e.g., emotional state interpretation; [5]), which leads to a more holistic approach to welfare management. Enrichment helps provide a more stimulating and complex environment more similar to that which the species would encounter in the wild and is now a common practice in modern-day zoos [6,7]. The main aim of enrichment is to increase in captivity the frequency and variety of species-specific behaviours that are conventionally displayed in the wild [7,8]. Enrichment can help captive facilities to improve both the psychological and physiological well-being of individuals and is an effective tool to increase the range and complexity of captive animals’ behaviours [4,9].

There are a number of ways in which zoos can try to enrich species to promote natural behaviours and stop abnormal behaviours. The most common form of enrichment is via feeding. Food-based enrichments provide the opportunity to look and search for food, allowing for expression of foraging behaviours, puzzle solving and exercise [10]. Other enrichment types involve sensory (visual, auditory, olfactory) and cognitive enrichment, which have both begun to receive more attention over recent years [6,11]. Auditory enrichment can range from playing noises from the wild (e.g., predator or prey noises) to different music types, such as classical or rock [8]. Visual enrichment often encompasses the use of colour, or technology, to provide novelty to the zoo environment [6,12]. Olfactory enrichment involves placing scents around an enclosure or presentation of scents to individuals [13]. Cognitive enrichment using technology has particularly increased over the last few years due to the increased availability of cheaper equipment [8]. When developing and using enrichment, it is important to consider whether the enrichment is suitable for the biology or ecology of the species [10], so enrichment should be considered at both species and individual levels [4].

To conclusively say enrichment is beneficial, certain standard measures are needed to assess the effectiveness of enrichment strategies on animal welfare [4] and ensure that management practices can be improved accordingly [6]. These measures are usually achieved by collecting behavioural and physiological data from individual animals [4,14]; however, so far, behavioural measures have been more commonly used compared to physiological measures [15,16]. The link between behavioural and physiological changes is tricky to validate and correlate in many studies, but it has been shown in a few cases (e.g., [14,17,18,19]). For example, one study [14] found that blue-and-yellow macaws (*Ara ararauna*) who expressed higher levels of behavioural activity and decreased abnormal behaviours had a lower concentration of cortisol in their faeces. Similarly, other authors [17] found that enrichment for African wild dogs (*Lycaon pictus*) increased their activity levels and was associated with decreased cortisol. Furthermore, a study [16] on cotton-top tamarins (*Saguinus oedipus*) showed that captive groups associated with higher frequencies of aggression had a higher concentration of cortisol.

Many studies have yielded valuable results showing that environmental enrichment can positively affect the behaviour of animals through increased activity, exploration and social behaviours, and decreased aggression or stereotypic behaviours [18]. For example, the use of various enrichments, such as food items and bubbles, significantly increased the exploration behaviour of harbour seals (*Phoca vitulina*) and grey seals (*Halichoerus grypus*), while reducing stereotypic swimming patterns [4]. Novel enrichments are more likely to have a noticeable impact on behaviour when compared to enrichments that have been used for a while as animals become habituated to them. This was illustrated in a study [19] on polar bear (*Ursus maritimus*) enrichment at the Bronx Zoo, where the authors observed that the use of novel toys (balls and floats) and a novel feeding device increased playing activity and decreased pacing when compared to previously used enrichments.

### 2.2. Captive Conservation Breeding

Zoos contribute to both in situ and ex situ conservation of endangered species through breeding programmes and by supporting reintroductions into the wild [2,20]. Notable conservation successes that have involved captive breeding include several primate reintroductions of captive-bred individuals, including golden lion tamarins (*Leontopithecus rosalia*; [21]), black-and-white ruffed lemurs (*Varecia variegata*; [22]), western lowland gorillas (*Gorilla gorilla gorilla*); [23]) and, most recently, pileated gibbons (*Hylobates pileatus*; [24]). Captive populations held in zoological facilities around the globe act as an important reservoir population for endangered species. They provide a safeguard against the extinction of species, with some currently only existing at all in zoos, such as the scimitar-horned oryx (*Oryx dammah*) and the Hawaiian crow (*Corvus hawaiiensis*) [2,20].

If the number of species continues to decline, there may potentially be a large increase in the number of captive breeding programmes needed [25]. For some species, it may be one of the only conservation options available until problems in species’ home ranges are addressed and overcome [25]. As such, it is important for conservation to implement an integrated approach by pairing both in situ and ex situ conservation to maximize the chances of success [26]. However, there are several primate species that have issues breeding in captivity, which is of concern due to their conservation status and continual decline in the wild [27]. For example, *Eulemur* species, gorillas (*Gorilla*), tamarins (*Saguinus*), gibbons (*Hylobatidae*) and howler monkeys (*Alouatta*) seem to breed less in captivity than in the wild [27]. Various studies have tried to understand the factors that may be contributing to this (e.g., [28,29]). Both mongoose lemurs (*Eulemur mongoz*) and black lemurs (*Eulemur macaco*) have shown higher rates of reproduction when breeding pairs are housed with either additional males or other breeding pairs, which suggests that the reproduction of these species is improved by the presence of conspecifics [30]. Likewise, a study on howler monkeys (*Alouatta caraya*) found that group composition also affected breeding success, showing that 38% of zoos housed the species in pairs despite no records of pairs being found in the wild [27]. Some primates may be hand-reared by keepers, which can cause problems with future reproductive success [28,29]. For example, a study on gorillas [28] found that those who were reared by their mothers were more successful in reproducing and rearing young compared to their hand-reared counterparts. As breeding programmes and endangered species management in captivity is already difficult from a physiological and social standpoint, it is important to understand what additional factors may influence reproductive success.

There are different techniques that researchers and zoos have tried to improve the success of breeding programmes, which are likely to continually improve as we gain an improved understanding about the reproductive biology of different species and increased technological capabilities [31]. For example, artificial insemination is a technique that has been widely used in zoos and is particularly important for species that only have a few remaining males left or when mating behaviour is not displayed in captivity [31]. Moreover, artificial insemination may also be useful in increasing the genetic diversity of captive populations [31]. Another important technique that can be used to improve reproductive success in captivity is the analysis of sex hormones in the urine and faeces of females to track oestrus [32]. This could help to better target and implement any breeding interventions that may be needed [32].

Additionally, mate choice is an important aspect of reproduction, even though this is rarely considered in captivity and pairings are most often based solely on studbook recommendations [1]. Mate choice is an important part of sexual selection and incompatibility between individuals can cause major issues. A study on captive giant pandas (*Ailuropoda melanoleuca*) [1] showed that females paired with preferred mates had higher rates of copulation and more births. Other authors [33] demonstrated that scents could be introduced in captivity to facilitate mate choice and significantly improve the chances for reproductive success in striped dunnarts (*Sminthopsis macroura*). Scents were also used to manipulate mate choice successfully in harvest mice (*Micromys minutes*) [34]. Although both studies used a relatively small sample size over a short period of time, they illustrate the potential of using scents to influence breeding of other captive species. Evidence (reviewed by [9]) indicates that olfaction has an important role in mate choice and sexual selection, with many implications for the success of species conservation.

Much of the focus in environmental enrichment studies is on the welfare of the individuals; little emphasis has been placed on how this, in turn, may affect other aspects of zoo biology, such as breeding [15]. The effects of enrichment on reproductive functions have been reported anecdotally over the years although there are few scientific studies that confirm this [13]. Thus, there remains a gap in knowledge of how enrichment may be developed or employed to help improve the conservation breeding of endangered species. Specifically, there are very few studies that focus on the impact of enrichment on reproduction as many only consider the effects on stress and physiology (e.g., [35,36,37]). However, there are various lines of evidence suggesting that it may affect potential reproductive success. As enrichment is designed to encourage individuals to display species-specific behaviour, including mating, it is likely to trigger or increase desirable behaviours associated with breeding [17]. Importantly, it is vital to ensure that species-specific socio-sexual behaviours are maintained in captivity so that individuals can breed unassisted after reintroduction to the wild to establish sustainable populations [27].

There have been a few laboratory-based studies showing a strong link between environmental enrichment and improved reproductive success. For example, a study [37] found that in zebrafish (*Denio rerio*), the number of eggs produced by breeding pairs was significantly higher in tanks that had a form of enrichment compared with control tanks. Similarly, another study [38] found that laboratory mice pups (*Mus musculus*) born in cages with enrichment often weighed more than those without enrichment and more of them survived to weaning age. Likewise, a further study also concluded that enriched mice pups had higher chances of survival and developed faster [39].

Even though the use of environmental enrichment is widespread across zoological facilities and has known benefits for welfare, the number of published studies on the relationship between enrichment and reproductive success is extremely limited (e.g., [13,17,35]). A study [40] on bobcats (*Lynx rufus*) at Jerez Zoo concluded that the females bred for the first time due the introduction of enrichment; however, they also indicated that there were changes to the groups and enclosures that could have been the main contributing factors to this. However, another study [35] showed that the ovarian function of both oncillas (*Leopardus tigrinus*) and margays (*Leopardus wiedii*) was disrupted when they were moved from an enriched enclosure to one with no enrichment. This link between enrichment and reproductive functions is further supported by a study on female African elephants (*Loxodonta africana*), which demonstrated that those with access to a higher diversity of enrichment had more regular ovarian cycles [36]. Conversely, another study [41] found that there were no changes in the prevalence of breeding behaviours for harbour or grey seals with or without enrichment.

There is currently only one major published study that aimed to look specifically at how novel enrichment could affect reproductive success in an endangered species. This study [36] investigated the effects of a novel auditory enrichment programme and its impact on a colony of northern bald ibis (*Geronticus eremita*) that were struggling to breed. The authors recorded calls of conspecifics during breeding season (reproductive cues), which were played back to the colony over the following two breeding seasons [17]. They found that breeding behaviours increased significantly, and more eggs were produced during the second season. This research provides an important example of the potential impacts of enrichment on conservation beyond just improving the welfare of individuals in captivity, while also highlighting the need for further studies investigating various novel enrichments that may ultimately increase the reproductive success of endangered species.

### 2.3. Enriching Primates

Primates have evolved many complex cognitive skills and behavioural patterns and have complex social lives, which often creates different challenges for these animals in captive environments compared to other non-social species [42]. Being unable to engage in these complex behaviours, or being housed in inappropriate social groups, can cause frustration and stress, which undermines their welfare. Some primate species may express high levels of stress or abnormal and stereotypic behaviours. Therefore, it is of the utmost importance to develop novel and sensory-stimulating enrichments aimed at primate species, particularly those that are currently underrepresented in enrichment programmes [42]. Despite a large proportion of enrichment articles focusing on primates, only a small minority of primate species were represented in these studies, highlighting the fact that bias towards certain species exists within orders and not just between taxa [35].

Enrichments for primates do not have to be complicated and many can be made from materials typically found in a zoo [43]. A study [43] validated increased activity levels of white-handed gibbons (*Hylobates lar*), brown lemurs (*Eulemur fulvus*) and Mona monkeys (*Cercopithecus mona*) in correlation with the introduction of a simple food-based enrichment device using bamboo and wire. They also showed it significantly decreased abnormal behaviours of brown lemurs when compared to the baseline [43]. Similarly, other authors [44] added boxes of food to a ring-tailed lemur (*Lemur catta*) enclosure and found it encouraged a small increase in active behaviours. Another study [10] showed that the use of suspended gum feeders for tamarins (*Saguinus* spp.) and marmosets (*Callithrix* spp.) significantly reduced the frequency of self-directed behaviours, which is an indicator of stress in primates. The use of music as an enrichment has also been suggested for primates, although results have so far been mixed. For example, the use of classical and rock music was found to significantly increase social behaviours and activity in gorillas [45], slightly increase activity and decrease abnormal behaviours in chimpanzees [46] and had no significant impacts on the behaviour of Moloch gibbons (*Hylobates moloch*; [47]). While most studies do show that enrichment has positive effects on captive primates, different enrichment types have differing effects on individuals and species (e.g., [10,41]). For example, a study on three different lemur species (ring-tailed lemur; red-ruffed lemurs, *Varecia rubra*; Coquerel’s sifaka, *Propithecus coquereli*) at the Duke Lemur Center (North Carolina) showed that each species reacted to three enrichments significantly differently based on their feeding ecology in the wild [7]. Additionally, further research [48] showed that abnormal and stress behaviours were significantly reduced across a group of captive chimpanzees during positive reinforcement training and that this effect was most pronounced among low-ranking individuals.

Cognitive enrichment has received much attention among primates, and particularly in great apes, and allows for healthy challenge and control in the zoo environment [11,16]. Some scholars [49] proposed the use of tools as a form of cognitive enrichment. They argued that wild chimpanzees spend a significant proportion of their time foraging, which often involves the manipulation or acquisition of food through tool use, and so provided captive chimpanzees with honey in bottles and gave them a few tools to extract it with [49]. They found that there was a significant reduction in inactivity in the group and an increase in social behaviours [49]. This has helped to illustrate the ways in which enrichment can be used to expand captive animals’ behavioural repertoire and to make their behaviours more comparable to those of their wild counterparts. However, some studies have found that individuals of social species monopolize the object or become frustrated when presented harder challenges, indicating the need for quantitative assessments of the benefits of each enrichment type prior to broad implementation [11,12,50].

## 3. Olfactory Communication

### 3.1. Neurophysiological Basis of the Olfactory Communication Systems

Animals detect and react to molecules deriving from the external environment, and all species perceive these chemical stimuli in the same way. Chemosensory receptor proteins are exposed to the outside world in the membrane of chemosensory nerve cells, frequently through a “window” in an otherwise impermeable skin or cuticle. The onset of an odour molecule (i.e., odorant, which may be any kind of molecule, from large proteins to small acids) is converted into a signal to the brain by binding to a chemosensory receptor protein (reviewed in [51]).

The combinatorial olfactory systems of animals can detect, discriminate, and distinguish between countless different molecules as different odours. This ability is conferred by the following features of olfactory systems: diverse chemosensory receptor proteins, with broad but overlapping specificities; expression of one receptor type per olfactory sensory neuron; and targeting olfactory sensory neurons sharing the same receptor to a collection point in the brain (i.e., glomerulus), one for each receptor type (reviewed in [52]). Olfactory receptor proteins are activated by multiple odorants, and then odorants activate multiple olfactory receptor proteins in different combinations. Most olfactory receptor proteins are broadly tuned so any odour molecule will stimulate a distinct subset of receptors and their associated glomeruli, giving a combinatorial code (i.e., “odour map”) representative of that odour molecule. The brain will then build up an olfactory picture of the external world from these, integrating the responses across the different glomeruli (reviewed in [51]).

Olfaction is a particularly important sense for many mammalian species, especially terrestrial species such as rodents and carnivores [53]. It is a key form of communication for both solitary and social species [54,55]. In mammals, different scent cues are managed by both the main olfactory system, whose function is to detect the volatile chemical signals, and the accessory olfactory system, which plays a key role in social communications [56]. Different mammalian species show diversity in the relative sizes of their olfactory systems and the number of functionally associated genes, which may be related to species’ sensitivity to odours [56].

Olfactory signals can come from a variety of sources, such as sweat, skin, specialized glands, urine, and faeces [57]. The chemical compounds found in odour secretions can be separated into volatile, semi-volatile and non-volatile organic compounds [57,58]. Volatile compounds are chemical compounds with a low molecular weight that can evaporate quickly into the atmosphere [58]. Non-volatile compounds are those with a higher molecular weight and do not readily evaporate, such as proteins. Semi-volatile compounds have higher molecular weights than volatile compounds but lower weights than non-volatile compounds, and their ability to evaporate readily is in-between those classified as volatile and those not. Volatile compounds are of particular interest in olfactory studies as they can be detected easily and quickly by the receiver [58,59]. Individual chemical signals may be very complex, containing a mixture of these different types of compounds [60]. Although compounds can be identified from olfactory signals, they are complex, sometimes containing hundreds of different compounds, of which some of the functions are not fully understood [60].

### 3.2. Olfactory Communication in Non-Human Animals

Olfaction is described as the sensory detection of chemical compounds originating from outside the body of the animal [61]. The olfactory system is phylogenetically the oldest sensory system and chemical signals are the oldest form of communication between animals [55,58,61]. Chemosensory systems (i.e., highly specialized sensory systems, of which taste and smell are prototypical examples; [62]) likely evolved early [54]. For this reason, they are found in most species, including reptiles, birds, and mammals. It is therefore surprising that, compared to the other senses, chemical communication in animals has long been understudied, with only recent work placing an emphasis on it (e.g., [9,58]). There has been an increase in published research relating to olfaction in recent years (e.g., [63,64]); however, there is still a lot currently unknown about olfactory communication in many vertebrate species. Despite this, it is established that olfaction is a key sense and has many important functions [13].

Olfactory communication has many advantages compared to visual, auditory, and tactile communication. Messages can be left in the environment for receivers without the presence of the signaller, which may help to avoid direct contact and limit physically aggressive encounters [9,65]. Unlike the other modes of communication, olfactory cues are unique in that they can persist in the environment [13]. Furthermore, odours are energetically costly to produce and reflect the condition of the individual; thus, they provide honest and reliable communication about the status of the individual [65,66]. Scent signals are not necessarily aimed at specific individuals but rather “broadcast” information to other individuals in the area, such as in the case of territory marking or advertising for a mate [59]. The functions and meaning of olfactory signals are, however, very difficult to interpret. Olfactory signals are usually extremely complex, and one signal may contain hundreds of different chemical compounds with varying functions [60]. Thus, they must be carefully deconstructed to infer their possible meanings [60,67].

Chemical communication, with chemical compounds involved in the chemical interaction between organisms called as “semiochemicals” [51], plays several crucial roles for different animal and plant species [54]. In this context, scent-marking behaviour is common in many terrestrial species and has a variety of functions, from marking home-range boundaries to signal that a competitor is in the area to establishing mother–offspring bonds [54,59,68]. Olfaction also plays a role in foraging (for example, by helping individuals to find the location of food sources and identifying their quality, which is advantageous in complex habitats where food may be dispersed over long distances or patchily distributed [54,56]) as well as in the detection of predators by prey species and vice versa [54].

Olfactory communication has also an important role in many social functions [65]. Many species release “pheromones”, which are defined as chemicals secreted by one individual that trigger a specific behavioural and physiological response from another individual from the same species [69]. Furthermore, scent signals can provide abundant information about the individual, including their sex, age, rank, and reproductive status, as well as individual and group identity [65,66]. There is also evidence to suggest that the major histocompatibility complex influences the volatile scents produced by mice, mandrills (*Mandrillus sphinx*) and humans (*Homo sapiens*) [59,70]. Signals and cues from odours are important for social communication, and this is emphasized in those species that have specialized glands and display scent-marking behaviours [64]. For example, studies have shown that higher-ranking or more dominant individuals display higher frequencies of scent-marking behaviours in sugar gliders (*Petaurus breviceps*; [71]), meerkats (*Suricata suricatta*; [72]) and mandrills [67].

### 3.3. Olfactory Communication in Primates

Primates are typically thought to be microsmatic, which means they are generally considered to have a poor sense of smell and instead mainly rely on visual and acoustic signals [73,74]. As such, primate olfaction is relatively understudied compared to other sensory modalities. It is believed that the reduced number of olfactory organs is related to the evolution of trichromatic vision in *Cercopithecidae* and apes [56,73]. However, strepsirrhine primates (“wet-nosed” primates) are known to rely heavily on olfactory communication and, accordingly, they display many genes associated with olfaction [73,74]. These species have more developed chemosensory organs and often show sexual dimorphism in scent glands [73,75]. Specifically, strepsirrhines and South American monkeys express far more functional genes related to olfaction and more developed chemosensory organs than African monkeys, Asian monkeys, and apes [73]. For example, research [76] showed that spider monkeys (*Ateles geoffroyi*) were able to discriminate between different odours at concentrations of just 1 ppm, demonstrating a level of sensitivity comparable to that of rodents and canines.

Both strepsirrhine species and South American monkeys rely on olfactory senses for different functions, such as foraging and communication of a variety of messages [68]. Most South American monkeys have specialized sternal or anogenital scent glands and produce odours via secretions from these glands [68,77]. For example, one study [78] was able to identify 162 chemical compounds from the scent marks of female common marmosets (*Callithrix jacchus*) and found that the relative abundance of compounds differed between individuals, indicating a role in identification. Furthermore, other authors [79] identified 110 chemical compounds from just two glands from monogamous owl monkeys (*Aotus azarae* and *Aotus nancymaae*), showing a large variation in the compounds based on the gland type, suggesting that they have different functions. Other studies also inferred different functions of olfactory cues based on differences between the sexes. For instance, one study [80] found that there were significant differences in scent-marking behaviour between the sexes in the moustached tamarin (*Saguinus mystax*) with females marking more frequently. Despite finding no difference in the frequency of marking between the sexes in black-tufted marmosets (*Callithrix penicillate*), there were significant differences in scent-mark deposition between males and females, suggesting different functions for marking in the two sexes [77].

Scent-marking behaviour is observed extensively in lemur species and serves various functions and can vary based on the species, age, and season [67]. Similarly, a number of studies have shown that scent-marking frequency tends to be higher in males than females (ring-tailed lemurs [81], red-bellied lemurs, *Eulemur rubriventer*; [82] and red-ruffed lemurs; [67]). The ring-tailed lemur exhibits complex olfactory behaviours and is often used as a model for lemur olfactory communication (e.g., [60,66,83]). Chemical analyses of lemur scents have revealed over 120 different volatile compounds that were found in different types of scent glands [57,65]. Despite the clear importance of olfactory communication in lemur species, most studies have only focused on the ring-tailed lemur (due to its commonality in captivity and complex olfactory behavioural repertoire), with other species underrepresented in the literature.

In contrast, there are only a few *Cercopithecidae* species that exhibit scent-marking behaviour, such as mandrills [68]. However, despite the general belief that olfactory behavioural patterns have lost significance in African and Asian primates, accumulated data suggest that they may be more important for these species than previously acknowledged [53,73]. For example, western lowland gorillas can discriminate between different odours, significantly increasing their investigation of a novel scent (almond or vanilla) [84]. Additionally, it has been shown that all great apes exhibit sniffing behaviours, and its frequency varies with species, sex, age, and context [55]. In all species, males display sniffing behaviours more often than females, with most sniffing occurring in non-social contexts. However, there are exceptions: male chimpanzees (*Pan troglodytes*) most often sniffed conspecifics [55], and mandrills, drills (*Mandrillus leucophaeus*) and olive baboons (*Papio anubis*) all displayed olfactory investigative behaviours during social foraging [85].

There is also more evidence emerging that suggests how *Cercopithecidae* and ape species can identify conspecifics via olfactory investigation (e.g., [86,87,88]). For example, one study [86] found that both male and female rhesus macaques (*Macaca mulatta*) investigated odours from individuals from outside their social group significantly longer than odours from within their group. In addition to this, another study [87] found that male Japanese macaques (*Macaca fuscata*) showed significantly more sniffing and licking behaviours when presented with female urine compared to a control odour. Likewise, further authors [88] found that when chimpanzees were presented with urine from conspecifics, the first response was an olfactory behaviour 83% of the time.

Chemical analyses of scent marks and body odours from African and Asian primates has helped to highlight the importance of olfactory signalling in these species. A total 77 chemical compounds were identified in odour secretions deposited by the scent-marking of male mandrills, with volatile chemical profiles conveying information about the sex, age and rank of the signallers [67]. In an analysis of axillary odours, 140 chemical compounds were detected in rhesus macaques [89]. The authors further analysed 21 of these compounds and found that there was variation in compound abundance and profile based on sex, kinship and group membership [89]. They also showed that the relative abundance of certain compounds was related to female rank [89]. Furthermore, another study [55] demonstrated via a chemical analysis of the body odour of great ape species that the chemical composition varied between species and individuals, which might underline the important social role that chemical communication could play in these species.

### 3.4. Olfaction and Reproduction in Primates

Olfaction plays a key role in reproduction. Scent signals provide information about the individual signaller [67,68], and scents are thought to provide honest indicators of mate quality. Therefore, scents are likely to play an important role in mate choice and successful breeding [68,74,75]. As chemicals produced by individuals depend upon the condition and health of the signaller, they likely reflect the physiological state of the individual [90]. There is a bias in the literature with studies on reproduction, chemosignals and individual quality focusing more on males than females [91]. Despite this, several studies have shown that chemical signalling from females can affect male physiology [91]. Females may release pheromones to induce certain sexual behaviours and physiological responses from males [91]. For example, the scents of new females can lead to increased testosterone production in marmosets (*Callitrichidae* spp.), macaques (*Cercopithecidae* spp.) and even humans [92,93,94].

Olfactory signalling plays an important role in advertising when a female is fertile, and this may or may not be associated with other sexual signals, such as visual sexual swellings or calls. The Graded-Signal hypothesis [95] states that exaggerated sexual swellings provide information about the probability of female ovulation and thus allow females to manipulate male behaviours. Further studies have shown that the olfactory behaviours of males change in response to females in primate species that display sexual swellings (e.g., [87,96]). A study [96] found that male Chacma baboons (*Papio ursinus*) displayed olfactory investigation towards females who had sexual swellings more often, suggesting a role of olfaction in advertising reproductive state. Other authors [87] proposed that olive baboons also use multimodal signalling to advertise female reproductive status, including olfactory cues. They found that the frequency of male olfactory investigations of females increased significantly during the females’ fertile phase [87]. Again, this indicates that olfactory signals are likely to play a role in providing information about female reproductive status.

Although some primate species show no visual signs of fertility, a broad range of studies have clearly shown a link between olfactory behaviours, the chemical composition of scents and reproductive status [74]. For example, one study [58] showed that male common marmosets increased their breeding behaviours when females were fertile despite a lack of visual cues, while the chemical composition of anogenital odours differed between fertile states, indicating an olfactory cue for reproductive status. Other studies have yielded similar evidence showing changes in scent-marking behaviours between breeding and non-breeding seasons in crowned lemurs (*Eulemur coronatus*; [97]), Milne-Edwards sifaka (*Propithecus edwardsi*; [53]) and ring-tailed lemurs ([98]). Ring-tailed lemurs and Milne-Edwards sifakas also showed differing chemical compositions of odours during breeding and non-breeding season [53,90]. Thus, a lack of visual or auditory cues does not correspond to an absence of olfactory cues [58]. As reproduction and parental investment are costly, especially for females, the advertisement of reproductive state will help an individual to improve their reproductive success [58,91].

Olfaction, furthermore, plays a role in parenthood, which is important for the successful raising of their offspring, and thus, reproductive success [99]. One of its most important functions is mother–infant recognition, which is crucial in establishing maternal bonds and individual identification [74,99,100]. A study [100] found that the odour profiles of female ring-tailed lemurs were significantly different before and during pregnancy, which further supports the role of olfaction in signalling reproductive status. It is also important for humans, as a study [101] found that 90% of mothers who participated were able to identify their babies’ clothes by smelling them. A more recent study [102] concluded that mothers could also identify toddlers from smelling their clothes, with the success of identification higher than chance rates. Their results also show that those women who were able to correctly identify their child were more successful in distinguishing between male and female odours [102].

Olfactory communication and behavioural repertoires are believed to have evolved under sexual selection [83]. For example, male ring-tailed lemurs exhibit a unique behaviour known as “stink fights” in which they anoint their tails and waft their scent towards competitors [83]. They also waft their anointed tails towards females (“stink flirt”), in what is likely a visual and olfactory display to advertise themselves as potential mates [83]. These behaviours are thought to be “honest” indicators of quality, as scents are costly to produce and reflect the physiological condition of the signaller [83]. Research [83] investigated this behaviour and found that higher-ranking males engaged in stink fights more often and much of the anointed wafting was directed towards females, suggesting it is likely a display of male quality.

## 4. Olfactory Enrichment

### 4.1. Scent-Based Enrichment in Captive Settings

The main goal of olfactory enrichment is to improve the welfare of animals in captive environments, but olfactory enrichment items can be used to manipulate the environment in other ways (reviewed in [103]). As studies have shown that habitat exploration can be influenced by the use of scent enrichment, this could be used to increase the visibility of animals to zoo visitors (e.g., [104,105]). Additionally, olfactory enrichment has shown to help increase activity and species-specific behaviours (e.g., [105,106]), which, in turn, could potentially provide a positive knock-on effect influencing the attitudes of the public towards zoo-housed endangered species and the frequency of zoo visits. Studies on domestic animals in shelters have shown that olfactory enrichment not only decreases stress but can help to increase positive behaviours and improve socialization (e.g., [107,108]). As scents often elicit behavioural and physiological responses, it is important to explore the use of olfactory enrichment to promote potential beneficial effects on reproductive success [17].

Olfactory enrichment offers several benefits that could be used by zoos. Scents are easy to apply as part of an enrichment programme, either via the introduction of the scents on cloths and other objects or their direct placement in the enclosure [8,109,110]. Furthermore, there are a number of different scents that could be introduced, ranging from natural (i.e., prey odours, relevant plant species) and anthropogenic (i.e., essential oils) scents to species-specific fragrances created in the lab [104,105]. Either scents from herbs and plants or animal odours from faeces could be easily obtained from areas within a zoo, providing a low-price and varied enrichment [17,106]. In addition to this, it is possible to introduce and mix scents in infinite ways, thus creating a dynamic and exciting enrichment schedule [104].

The effects of olfactory enrichment have been tested by a number of studies on domestic, farm, laboratory and zoo-housed species (e.g., [111,112]). For instance, olfactory enrichment has been studied in both dog (*Canis lupus familiaris*) and cat (*Felis catus*) rescue shelters as a tool to reduce stress and encourage desirable behaviours (e.g., [108,110,112]). Rescue shelters are often stressful environments, and it is therefore important to find ways to improve the welfare of individuals. A study [108] exposed domestic dogs to a variety of essential oils and blends of oils and found that some scents increased frequencies of behavioural indicators of relaxation while others decreased behavioural indicators of stress such as pacing and over-grooming. Similarly, another study [110] found that the introduction of cloths scented with vanilla, coconut and ginger significantly decreased vocalizations associated with stress in dogs housed in a rescue centre. A study on domestic cats [107] showed that almost all individuals showed a positive response, including increased play behaviours and decreased stress behaviours, when exposed to a few scented plants. Further studies [107,113] found that olfactory enrichment involving catnip significantly increased play behaviours, which is highly desirable for facilitating adoptions.

Generally, in zoo-housed animals, studies have found that scent enrichments could be effective at increasing active and natural behaviours, such as enclosure exploration and scent marking, and might contribute to improving welfare of various species (e.g., [104,105,106]). For example, a study [105] introduced various naturally occurring (e.g., kelp and sardine oil) and biologically relevant (e.g., vanilla and cinnamon) scents to a group of Californian sea lions (*Zalophus californianus*) and found that these scents significantly increased enclosure utilization and reduced stereotypic swimming patterns. Similarly, a study on Rothschild giraffes (*Giraffa camelopardalis rothschildi*) [104] showed that habitat exploration and activity increased with the introduction of a scent-based enrichment. Other authors [114] found that the activity and exploration of black-footed cats (*Felis nigripes*) significantly increased during olfactory enrichment, while both resting and standing behaviours decreased. Moreover, a study on cheetahs (*Acinonyx jubatus*) and tigers (*Panthera tigris*) [115] found that stereotypic pacing behaviour was significantly decreased in the presence of a hay ball with cinnamon. Additionally, further studies [17,116] showed that the activity of African wild dogs was significantly increased via the introduction of scent-based enrichment.

However, some studies showed findings that were less clear or indicated that scent enrichment had little effect (e.g., [6,117]). For example, a study [117] found that the introduction of scents, including prey odour, had no significant effects on the behaviour of a group of captive meerkats, despite olfaction being used extensively by the species. However, as acknowledged by the authors, there was no enrichment-free control used in the study to establish a baseline of behaviour without presentation of a novel object. It is, therefore, difficult to conclude that the enrichment was not effective [117]. Furthermore, another study [107] found that 65% of tigers did not respond to an olfactory enrichment with the rest showing only partial behavioural responses, in contrast to other studies on big cats (e.g., [106,115,118]). Similarly, other authors [108] found that a decrease in faecal cortisol levels was exhibited by domestic dogs despite differing levels of exposure to essential oils.

As a few studies have indicated [11], the success of novel olfactory enrichment programmes seems dependent on both the delivery of the scent and the type of scents used. Most studies used spices or essential oils, rather than focusing on natural or biological scents that may be more meaningful to the species. Many scents, such as lavender, were chosen based on their effectiveness in humans or domestic animals, but this may not necessarily be appropriate for all species [103]. As with all types of enrichment, the biology of the species should be considered, and its effectiveness should be continually monitored to inform best practices. For example, several studies have suggested that the use of either natural prey/predator odours or scents from conspecifics should be researched further (e.g., [8,119]). Further studies have suggested that the use of diffusers as a delivery method may be more effective (e.g., [11]). Additionally, other authors have indicated that scents could be used in a number of combinations and introduced randomly in order to continue to add novelty to the enrichment programme and avoid the problem of habituation (e.g., [106]).

There have been several studies that have used prey or predator odours to test the responses by individuals (e.g., [118,120]). For example, the introduction of faeces from prey to captive lions (*Panthera leo*) has been shown to significantly increase their activity [118]. On the other hand, several studies have shown mixed effects when introducing predator odours to captive animals [8]. For example, cotton-top tamarins reacted with high anxiety behaviours when exposed to predator faeces [121], while exposure to wolf (*Canis lupus*) odours elicited no response from horses (*Equus ferus caballus*) [120]. A study [113] also found that domestic cats showed little interest in cloths that were impregnated with rabbit (*Lagomorpha* spp.) odours. Another study [17] found that African wild dogs only showed an increase in activity when introduced to natural prey scents rather than other scent types, suggesting that it had more meaning and thus a greater response from the animals. In contrast to this, other authors [116] found that both natural (blood trail from prey) and non-natural (lemon) scents led to a significant positive response in African wild dogs. As animals are unlikely to face prey or predators in a captive environment, these may then be less relevant to captive-born individuals than their wild counterparts. However, these studies still show promise and demonstrate a need for further investigation into what constitutes a relevant odour and how this can be used to create a more stimulating environment for captive animals.

### 4.2. Scent Enrichment for Primates

The overall effects of olfactory enrichment on primate species are currently unclear and understudied. A study on Moloch gibbons [119] found that olfactory enrichment did significantly increase the frequency of species-specific behaviour, although individuals’ interest in the olfactory enrichment decreased rapidly after the first day, which was undesirable. By contrast, another study [8] tested the effects of olfactory enrichment on ring-tailed lemurs at Birmingham Wildlife Conservation Park (UK) and found no significant effects on individuals’ behaviour in the presence of the odours; however, this was a relatively small study (eight individuals in one zoo) and only one scent used was ecologically relevant to the species. Likewise, scents were also shown to have no significant effects on the behaviour of gorillas at Belfast Zoological Gardens (UK), but again only essential oils with little ecological relevance to the species were used [109]. Thus, it is important to consider the ecological/biological relevance of enrichment to the species as this is likely to affect the results.

Our preliminary study [65] on the effects of essential oils, such as benzoin, lavender, and lemongrass, show that these oils may work as scent enrichment to decrease the stress levels of zoo-housed primates across the major lineages. This is particularly the case of social primate species where odour plays a crucial role, such as red-ruffed lemurs. Specifically, after exposure to a series of essential oils, red-ruffed lemurs spent less time engaging with reassurance-derived social interactions and exhibited lower rates of stress-related behaviours but increased levels of faecal glucocorticoid concentrations, which could be associated with an increase in activity by individual lemurs.

Different studies (e.g., [8,109,119]) suggested that an emphasis should be placed on the use of ecologically or biologically relevant odours as a form of olfactory enrichment; however, there remains a significant lack of research in regard to this. Most scents currently used in these studies are essential oils, spices, or herbs. One study [8] used a prey odour (meal worms), but the authors stated that the ring-tailed lemur troop was fed these regularly anyway and thus it had no novelty. All these studies [8,109,119] suggest that prey, predator, and conspecific odours should be considered and studied as a form of enrichment.

Despite some authors highlighting that using olfactory cues can improve the breeding success of captive animals by facilitating mate choice (e.g., harvest mice, [15]; striped dunnarts, [9]), this topic is still understudied. While neither of these studies were focused on enrichment and the odours presented were not meant as such, they both provide evidence suggesting that olfactory enrichment could influence breeding behaviours as well as welfare. Specifically, these studies show the need for further exploration of biological relevant scents as novel enrichment programmes.

## 5. Conclusions

Studies on olfactory enrichment are less frequent than those on other enrichment types, and additionally show a bias towards big cat species [13]. In addition, there is mixed and conflicting evidence regarding the benefits of scent-based enrichment for captive animals’ well-being [8]. However, several studies (e.g., [116,118,120,121]) showed promise and demonstrated a need for further investigation. Future work should focus on what constitutes a relevant odour by considering the ecological/biological relevance of olfactory enrichment to the species, and how this can be used to create a more stimulating environment for captive animals. Thus, future research would need to focus on further exploration of biological relevant scents as novel enrichment programmes and how to use such novel enrichments to encourage breeding behaviours of endangered species in zoos. While the link between olfactory cues and the breeding success of captive animals is still understudied, there is accumulating evidence (e.g., [9,15]) suggesting that olfactory enrichment could influence mating behaviours as well as welfare status. Olfactory-based enrichment may benefit endangered species included in ex situ conservation initiatives [9,17], as shown by our preliminary findings [65,122,123] on newly designed scent enrichments, mimicking female fertile odour signals to trigger male mating behaviour and hopefully improve reproductive success in endangered zoo-housed lemur species.

## Data Availability

Not applicable.

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
