# Peer review of "The Scent Enriched Primate"

_animals, 2023, doi:10.3390/ani13101617_

Round 1
Reviewer 1 Report
Dear authors,
I found your review of interest and that could be useful to organize and give clarity and perspective to previous studies in scent enrichment in primates.
However, I found some aspects, that I suggest to manage.
General points
1. I would suggest to address clearly which is the aim of the review, because otherwise we don't understand where are we going through the different parts.
This aim could be followed also by brief schema of the logic in the presentation: olfactory communication-->primates...
2. I would suggest focussing the review in a more specific issue (scent enrichment in primates) and use more briefly the other parts (breeding in zoos, pp 10-13 general enrichment pp 6-7).
3. I would suggest adding some clear insights about olfactory enrichment in primates which summarize your findings in the literature. For example, a summarizing table. A figure showing different enrichment methods in primates would be also helpful for clarity.
Minor comments
L93 Please include the definition of semiochemical for chemical communication. eg: Wyatt et al 2003-2014.
L158 Please could you add any insight about why is this species overrepresented.
L175 I suggest to erase "presented" in "presented odours" for clarity
L223 I suggest no "visual signs" instead of "no visible".
L291 I suggest: "the main aim of enrichment in captivity is..."
Author Response
General points
1. I would suggest addressing clearly which is the aim of the review, because otherwise we don't understand where are we going through the different parts.
This aim could be followed also by brief schema of the logic in the presentation: olfactory communication-->primates...
R: We have added a preamble with aim and outline of the review.
2. I would suggest focussing the review in a more specific issue (scent enrichment in primates) and use more briefly the other parts (breeding in zoos, pp 10-13 general enrichment pp 6-7).
R: We have substantially amended the review by focusing it on scent enrichment in primates while shortening the other sections.
3. I would suggest adding some clear insights about olfactory enrichment in primates which summarize your findings in the literature. For example, a summarizing table. A figure showing different enrichment methods in primates would be also helpful for clarity.
R: We have added a summarizing table to the section Enriching Primates.
Minor comments
L93: Please include the definition of semiochemical for chemical communication: e.g., Wyatt et al 2003-2014.
R: Included.
L158: Please could you add any insight about why this species is overrepresented.
R: Added.
L175: I suggest erasing "presented" in "presented odours" for clarity.
R: Erased.
L223: I suggest no "visual signs" instead of "no visible".
R: Amended.
L291: I suggest: "the main aim of enrichment in captivity is..."
R: Amended.
Reviewer 2 Report
While I appreciate the desire to publish a review on primate-specific olfactory enrichment, I feel as though (as written) the current review does not actually offer enough about primates and feels fairly disjointed. I think there is a lot of promise, however, and I look forward to reading the next version.

Author Response
While I appreciate the desire to publish a review on primate-specific olfactory enrichment, I feel as though (as written) the current review does not actually offer enough about primates and feels fairly disjointed. I think there is a lot of promise, however, and I look forward to reading the next version.
General Comments
- Please read Clark & King (2008): A Critical Review of Zoo-Based Olfactory Enrichment found in Chemical Signals in Vertebrates, 11. A lot of your review has already been discussed in this one. If you could either tailor your review to highlight what has been published since the previous review article, or focus more on the specifics of primates, it would provide a more useful review article to those who want to use it.
We have substantially amended the review, which now focuses more on the specifics of primates.
- My biggest concern is the flow of the paper. At the moment it reads like two separate reviews – one on the importance of including the olfactory system in enrichment management and the other on how enrichment, as a whole, contributes to reproductive success. Please be very deliberate on your editing to try and link these two reviews together as this is what differentiates your review article from some already written.
- A suggestion would be to start with the enrichment to conservation link / your background information on enrichment as a whole and then move into the specifics regarding olfactory enrichment.
We have amended the review by following this suggestion.
- Additionally, re-working the abstract / simple summary to truly reflect the review. At the moment, it is setting up the reader to think that the majority of this review is on how enrichment is linked to reproductive success – it doesn’t provide enough emphasis on scent-based enrichment, nor on primates.
We have re-worked both the abstract and the simple summary.
- Re-check the citation style – Animals is very particular about how they cite. Fix it now so you can adjust any flow issues that come up.
We have revised the in-text citation style.
- Please create a small section that goes into the physiology of the olfactory system – at the moment, you dive into the review without any background on the system itself. I think it will make this paper much more accessible to people outside of animal experts (especially managers who may or may not have a background in animal physiology and are frequently making welfare decisions).
We have added a short section covering the physiology of the olfactory system.
- Replace “feeding” enrichment with “food-based enrichment” as sometimes the enrichment program doesn’t actually lead to consumption of the food items.
Replaced throughout the manuscript.
- Also make sure there is a comma after each “e.g.” (should read “e.g.,”).
Done throughout the manuscript.
- You need to include a stronger concluding section that links all the things you are discussing. It should mention how olfaction-based enrichment will benefit species included in ex situ conservation work – with a specific and detailed assessment on primates. That should help with tying everything together.
We have revised the conclusion section according to this comment.
Simple Summary & Abstract
Line 13: “…face of a global diversity” – catastrophe? Decline? Needs another word here.
Replaced with “decline”.
Line 16: remove “on” after “impact”
Removed.
Line 17: Break up sentence. “…success of individuals. Consequently,…”
Done.
Line 18: what type of negative effects? Physiological? Behavioral?
Amended (behavioural, physiological, and cognitive).
Line 21: Are scent-based enrichments less used? Might be worth stating that they are not utilized
despite their promise in multiple research studies.
Done.
Line 22 – 24: This sentence feels like an add-on and irrelevant here. Either needs to be worked
into the general enrichment background information at the beginning of the abstract / simple
summary or removed.
Removed.
Lines 25-39: Similar edits to the simple summary – but for a review on scent-based enrichment,
there are only two mentions of the olfactory system and scent-based enrichment in the entire
abstract. This is setting up the reader to think that this is a review of enrichment as a whole, not a
specific type of enrichment.
Amended.
Keywords: Shorten or split the first one.
Shortened.
Olfactory Communication
I think it would help the reader a lot to have a short summary of the physiology of the olfactory
system – before “Olfactory Communication” would be an ideal place to put it.
Added.
Line 48: remove the comma and replace “while” with “and”
Removed.
Line 49: provide a quick definition of chemosensory systems
Done.
Line 54: “…relating to olfaction over recent times…” – please add some examples of recent
publications (e.g., Vaglio et al 2021 or LaDue & Schulte 2021)
Added.
Line 55: remove “now”
Removed.
Line 56: replace “plays” with “has”
Replaced.
Line 58: suggested re-write – “…environment for receivers without the presence of the signaler, which may help to avoid…”
Done.
Line 60: should be “encounters”
Amended.
Line 63: replace “depend upon” with “reflect”
Replaced.
Lines 71 – 92: Would be better if moved to the background section on the olfactory system that I
recommended.
Done.
Olfactory Communication in Primates
Need to introduce the concept of primates having reduced olfactory organs earlier.
Done.
Line 118: comma after “As such” and remove “still”
Added.
Line 119-120: remove from “despite” to “primate species”. Irrelevant info.
Removed.
Line 126-128: Flip the subject of this sentence. Talk about the differences in relation to the
Strepsirrhines and SA monkeys since you spent the previous paragraph discussing them.
Amended.
Move lines 130-133 (“For example… rodents and canines”) to right after the first paragraph in this section.
Done.
Line 139: replace “discovering” with “showing”.
Replaced.
Line 152: remove “the most heavily studied of all strepsirrhine species”.
Removed.
Lines 167-170: re-write to condense. E.g., “…with most sniffing in non-social contexts. However, there are exceptions: male chimpanzees (P. troglodytes) most often sniffed conspecifics, and mandrills, drills(sp), and olive baboons (sp) all displayed olfactory…”
Revised.
Line 175: Insert “social” before “group significantly”.
Done.
Lines 193 – 202: I would focus on non-human primates as human olfaction is a gigantic topic that
has actually been well-researched. If you make it clear that you’re interested in reviewing nonhuman primates, I think it would make your manuscript stronger.
Amended.
Olfaction and Reproduction in Primates
Line 207-209: Is it only males that produce odours that reflect their condition?
No… Amended.
Line 220: remove extra space in citation.
Removed.
Line 221: change “must” to “are likely to”.
Changed.
Line 233-234: rewrite. E.g., “…visual or auditory cues does not correspond with an absence of
olfactory clues.”
Amended.
Lines 259-266: Move this paragraph and integrate it with the beginning of the section when you start discussing how males produce odours that reflect their condition.
Done.
Environmental Enrichment in Zoos
Shorten this whole section! So many reviews already exist of this that you can significantly
shorten this entire section. Also, move it to the very, very beginning of your review.
We have substantially shortened and reworked the section according to this comment.
I recommend starting at “Environmental enrichment is widely implemented…”
Done.
Line 286-287: Interesting, a lot of welfare research integrates immeasurable aspects (such as
emotional state interpretation – see Wemelsfelder & Farish 2004) to develop a more holistic approach to welfare management.
Amended.
Line 306: remove “This type of enrichment…different species” – interesting but not really relevant.
Removed.
Line 316: Your statement about the link between behaviour and physiology is too general. This is actually one of the hardest things to validate and correlate. For every example where there is a
clear correlation between behaviour and physiological changes (especially hormonal changes)
there are at least two that show there is no correlation.
Amended.
I don’t think you need lines 325 – 333.
Deleted.
Enriching Primates
Line 355: “Skilles” should be “skills” and remove “so” and “they”.
Amended.
Line 356: replace “show” with “have”.
Replaced.
Line 357: insert “non-social” after “other”.
Done.
Line 364: This review was published almost 8 years ago. Do a quick search and make sure your (and their) numbers are up-to-date.
We have now amended this sentence.
Line 366: re-write. E.g., “…produced a feeding enrichment device utilizing bamboo and wire that corresponded with increased activity levels of white-handed…”
Revised.
Line 368: random “;” after “Cercopithecus mona”.
Deleted.
Line 388: you should introduce the concept of healthy challenges for zoo-animals, otherwise this sentence isn’t as positive as I think you mean it to be.
Done.
Line 390-391: re-write. E.g., “…significant proportion of their time foraging which often involves the manipulation or acquisition of food through tool use…”
Revised.
Olfactory Enrichment
Line 401: Again, this review is now 12 years old. Make sure information is still accurate.
Done.
Line 402 – 403: This sentence needs to be both re-written and moved to the very, very beginning of your review. You must define olfactory stimulation as soon as possible.
Done.
Line 410: Is this always true? Your previous statements that indicate scents can linger in the
environment vaguely contradict this.
We have now deleted this sentence.
Line 413: add “species-specific” before “fragrances created in the lab” Lines 418-419: remove each instance of the word “animals”.
Added.
Line 434: remove “indeed”.
Removed.
Line 436: remove “ultimately”.
Removed.
Line 438: cinnamon and vanilla are naturally occurring scents. Do you mean “biologically
relevant”?
Yes… Amended.
Line 459-461: re-write. E.g.,”Uccheddu et al 2018 found a decrease in faecal cortisol levels was
exhibited by domestic dogs despite differing exposure to essential oils.”
Revised.
Line 462: remove “for” and change “implementation” to “success”.
Removed.
Line 463: change “it is important to consider” to “are dependent on”.
Changed.
Line 646: remove “have”.
Removed.
Line 467: remove “However”.
Removed.
Line 485: replace “when compared to” with “rather than” and add a comma after “scent types”.
Replaced.
Line 488: replace "had” with “led to” and remove the capital “W” from “Wild”.
Replaced.
Line 489: You need a phrase like “than their wild counterparts” or something to indicate that captive-bred species specifically are less impacted by natural odours.
Amended.
Line 493 – 505: Move whole paragraph earlier.
Done.
Like 494: re-write. E.g., “but olfactory enrichment items can be used to manipulate the
environment in other ways.”
Revised.
Like 498: remove extra space in citation.
Removed.
Line 502: remove “which may result being important for adoptions”.
Removed.
Line 503: remove “then” Olfactory Enrichment for Primates.
Removed.
Line 521: replace “knowledge” with “research” Line 522: replace “remains largely focused on” with “are”.
Replaced.
Line 523: what type of lemur?
Amended.
Conservation Breeding
This section is where it really starts to feel like a whole new paper.
You can SIGNIFICANTLY edit this shorter so that you have more room to really elaborate on
Primates.
We have substantially shortened this section.
Line 531: Any updated numbers than from 2013?
We have now deleted this sentence.
You need to make your connection between enrichment and conservation breeding programs
much clearer and sooner.
Done.
Line 579: 66% success rate in reintroductions is actually much higher than it historically was and
not too bad.
We have now deleted this sentence.
Personally, I think you can merge the first seven paragraphs of this section into 2, or 3 maximum,
paragraphs and start the main body of this section with “There are several primate species that
have issues breeding in captivity…”
Done.
Line 606: replace “if” with “when”.
Replaced.
Line 615-616: This little paragraph feels like the intro to a new section. It also feels like a good time to reintroduce enrichment.
Done.
Line 650-651: there have been more papers published on this link (e.g., Meikle et al. 2020) – also
your very next sentence actually shows that there were significantly MORE research articles than
those reported by Alligood and Leighty so I would just remove this sentence in general.
Removed.
Line 656: need an example citation for enrichment effects on stress and physiology.
Added the citation.
Line 658: remove “it could be used” Line 661” remove “in order”.
Removed.
Line 696-704: considering your paper is on olfaction enrichment in primates I think you really need to elaborate this last paragraph and include examples on how scents could assist primates.
Even if there aren’t formal research papers out there with this specific link, use the previously cited papers in the two paragraphs above to argue why scent-based enrichment is so important and exciting for primate conservation.
We have expanded this paragraph and included relevant examples.
Round 2
Reviewer 2 Report
All in all, this is MUCH better than the previous version. It may look like I have added a lot of comments but the majority of them are centered around sentence structure / word choice. The content was much clearer and easier to follow.
One big thing I recommend doing is going back and making sure the overall tense (past, present, or future) is consistent. Sometimes you use past-tense ("studies have shown") and other times you use present ("lemurs are changing behavior"). Just make sure it is consistent throughout the review.
Also, go through and triple check you have added in the full scientific name of each species at the FIRST instance they come up. At other times throughout the review you can use the condensed version (e.g., A. jubatus).
Otherwise, much much better.
Please see attached .pdf for specific recommendations / comments.
